# Strategic Planning for Tuberculosis Control in the Republic of Fiji

**DOI:** 10.3390/tropicalmed4020071

**Published:** 2019-04-24

**Authors:** Romain Ragonnet, Frank Underwood, Tan Doan, Eric Rafai, James Trauer, Emma McBryde

**Affiliations:** 1Department of Medicine, the University of Melbourne and Department of Public Health and Preventive Medicine, Monash University, Melbourne 3004, Australia; Romain.Ragonnet@monash.edu; 2Ministry of Health and Medical Services, Suva, Fiji; dr.funderwood2011@gmail.com; 3Australian Institute of Tropical Health and Medicine, James Cook University and Department of Medicine, University of Melbourne, Melbourne 3000, Australia; tan.doan@uqconnect.edu.au; 4Ministry of Health and Medical Services, Suva, Fiji; Eric.rafai@govnet.gov.fj; 5Department of Public Health and Preventive Medicine, Monash University, Melbourne 3004, Australia; james.trauer@monash.edu; 6Australian Institute of Tropical Health and Medicine, James Cook University, Townsville 4811; Australia

**Keywords:** tuberculosis, health policy, disease modelling, simulation, epidemiology, public health

## Abstract

The tuberculosis (TB) health burden in Fiji has been declining in recent years, although challenges remain in improving control of the diabetes co-epidemic and achieving adequate case detection across the widely dispersed archipelago. We applied a mathematical model of TB transmission to the TB epidemic in Fiji that captured the historical reality over several decades, including age stratification, diabetes, varying disease manifestations, and incorrect diagnoses. Next, we simulated six intervention scenarios that are under consideration by the Fiji National Tuberculosis Program. Our findings show that the interventions were able to achieve only modest improvements in disease burden, with awareness raising being the most effective intervention to reduce TB incidence, and treatment support yielding the highest impact on mortality. These improvements would fall far short of the ambitious targets that have been set by the country, and could easily be derailed by moderate increases in the diabetes burden. Furthermore, the effectiveness of the interventions was limited by the extensive pool of latent TB infection, because the programs were directed at only active cases, and thus were unlikely to achieve the desired reductions in burden. Therefore, it is essential to address the co-epidemic of diabetes and treat people with latent TB infection.

## 1. Introduction

Tuberculosis (TB) remains endemic in the Republic of Fiji (hereafter referred to as Fiji) with an estimated incidence of 59 new cases per 100,000 population in 2016 (95% confidence interval [CI] 45–75) [1]. The fight against TB in the South Pacific Ocean region in general and in Fiji in particular is complicated by local challenges, including a substantial part of the population living in isolated rural areas and high rates of type 2 diabetes mellitus (T2DM) [2]. Fiji comprises more than 100 inhabited islands, most of which are remote from the main two islands (Viti Levu and Vanua Levu) where the TB clinics are located, constituting a challenge for case detection and patient management. The TB epidemic in Fiji is significantly driven by T2DM, which is on the rise and affects around 16% of the Fijian population aged 25 years and above [3]. T2DM is estimated to increase the risk of developing TB disease around threefold [4], making this comorbidity an important concern for the national TB control program (NTP). By contrast, HIV is not a major driver of the TB epidemic in Fiji, due to its very low prevalence in the country [5]. 

Directly observed therapy short course (DOTS) has been the standard of TB care in Fiji since 1997, with most patients being hospitalised at the start of the treatment, but later receiving community-based treatment [6]. This has contributed to sustained high rates of treatment success over the last two decades, and to minimal multidrug-resistant TB (MDR-TB) in the country. Although the first ever case of MDR-TB was identified in 2016, there is no evidence of transmission within the country.

Since 2010, the Global Fund to Fight AIDS, Tuberculosis, and Malaria (GFATM) has been the major donor to the Fiji NTP, contributing approximately half the country’s total funding for TB. This funding has enabled strengthened active case-finding (ACF) activities and the implementation of GeneXpert for rapid diagnosis and the identification of rifampicin resistance. However, following the updated classification of Fiji as an upper–middle income (previously lower–middle income) country by the World Bank in 2012, Fiji has become ineligible for future GFATM funding, and the final grant from GFATM ended in 2017. With this return to almost fully domestically funded TB services, it is critical to strengthen and improve the efficiency of TB programs in Fiji. This is particularly important if the country is to achieve the NTP’s targets of 80% reduction in TB incidence and 90% reduction in TB mortality by 2030, which are in line with the Sustainable Development Goals (SDGs) and the End TB Strategy.

Transmission modelling is increasingly used to make predictions about the future trajectory of the TB epidemic, as well as effectiveness and costs associated with planned control interventions. It has become a critical tool for policy makers in the cyclical process of evaluating current programs, setting future priorities, and developing funding requests to donors. We previously developed and described the AuTuMN software platform to guide country-level TB decision making and strategic planning [7]. Here, we describe the application of this tool to TB control in Fiji.

## 2. Materials and Methods 

### 2.1. General Approach

The approach to development of the AuTuMN software platform is described in detail in Trauer et al. [7]. Briefly, the software assists countries with TB decision making and strategic planning by providing (i) predictions of the future TB epidemic trajectory based on current epidemiology and programmatic responses, (ii) the likely epidemiological and economic impact resulting from future changes to the programmatic response, and (iii) the cost-effectiveness of TB programs. The full details of the model are described in the Appendix A and summarised as follows. Given the slow moving and complex nature of the TB epidemic, we aimed for consistency with historical TB dynamics in Fiji for many decades into the past, and allowed a high degree of model complexity. To achieve this, we first loaded the publicly available data for birth, death, Bacillus Calmette–Guérin (BCG) vaccination and TB-related programmatic measures and fit functions mapping parameter values to time for these quantities using polynomial spline functions (Appendix A).

### 2.2. Background Demographics

Patients are first born into the model with the time-variant birth rate fit to World Bank data (Appendix A) and proportions of births vaccinated and unvaccinated are split according to the vaccination coverage data from the World Health Organization (WHO) or the United Nations Children’s Fund (UNICEF) (Appendix A). Population-wide natural mortality was also parameterised to data from the World Bank (Appendix A) and was applied to all the compartments. All the compartments were stratified by age, with age groups being: under 5 years, 5 to 15 years, 15 to 25 years, and 25 years and above. The proportion of the population with T2DM was considered as a time-variant parameter and applied to the oldest age group only (Appendix A).

### 2.3. Model of TB Transmission and Progression

The force of infection was calculated as the number of infectious persons, including undiagnosed persons, missed false-negative presentations who had yet to seek additional care, and patients in very early stage treatment. All the paediatric (<15 years old) and smear-negative TB patients were considered to have markedly reduced infectiousness compared to adults. The force of infection was further modified according to the susceptibility/immunity status of the susceptible group, with BCG-vaccinated children retaining immunity from vaccination until 15 years of age, and latently infected persons also having a markedly reduced risk of infection (Appendix A). The relative immunity attributable to previously treated disease episodes is highly uncertain, and so was varied as a calibration/uncertainty parameter.

Following infection, the latency period to active TB disease was simulated as two sequential compartments, with infected persons initially entering a high-risk early latent state before transitioning to a lower risk late latent compartment [8,9]. The proportion of infected persons progressing to active TB disease from early latency is greater for children than adults, and each new episode of active TB disease is distinguished according to smear status into smear-positive, smear-negative pulmonary, and extrapulmonary using country-specific notification data. Persons with T2DM have an approximately threefold increased risk of progression to active TB disease following infection (Appendix A) [4]. 

An untreated episode of active TB was assumed to last for around three years (although uncertainty was included around this parameter), with a greater case fatality rate for smear-positive active TB than for smear-negative and extrapulmonary disease (Appendix A) [10]. 

### 2.4. Simulation of the Health Care System

The rate of detection of active TB patients by the health care system was set such that the proportion of cases that were detected—rather than reaching an outcome as a consequence of the natural history of their disease episode (i.e., spontaneous recovery or death)—was equal to the reported time-variant case detection rate (Appendix A). Smear-positive and extrapulmonary patients commenced treatment after a shorter delay than smear-negative patients, since their diagnosis relies on clinical judgement and sputum smear rather than culture. Treatment outcomes were divided into success, death, and/or unfavourable outcomes other than death, with treatment rendering the patient non-infectious after 10 days and lasting for six months in total. The probability of reaching each of these three outcomes varied with time and differed according to treatment history (Appendix A).

### 2.5. Automatic Calibration and Uncertainty

We used a Metropolis algorithm to calibrate the model to align with the TB incidence estimated by the WHO for the years 2010 to 2016, by adjusting five important but uncertain parameters: the effective contact rate, the TB epidemic starting time, the relative risk of reinfection in previously treated individuals, the case fatality ratio for untreated TB cases, and the duration of untreated active disease. A Metropolis algorithm is a Markov chain Monte Carlo method that enables sampling from the posterior distribution of the model parameters [11]. That is, it identifies the model parameter sets that are best able to replicate the trends observed in the data. Details about the Metropolis algorithm used in this study are available in the Appendix A. To match reported mortality rates, it was necessary to incorporate a parameter reflecting the proportion of deaths among TB patients who never reached the health system that contributed to the reported mortality. Comparison to prevalence and trends in reported mortality over time were assessed to validate calibration, but were not part of the calibration algorithm. The model runs accepted by the Metropolis algorithm (which were associated with the posterior parameter distributions) were used to assess uncertainty around the reported epidemiological outputs, and the parameter set with the greatest likelihood from 10,000 iterations was used to project the future epidemiology under scenario conditions. 

### 2.6. Interventions

In addition to the baseline conditions in which all the time-variant parameters were maintained at their 2016 values, we developed a set of six programmatic interventions in collaboration with the Fiji NTP (Table 1). These interventions were intended to reflect the real-world programmatic changes that are currently under consideration by the country stakeholders. The interventions were modelled by changing the relevant model parameters from their baseline values, using the continuous scale-up function to increase the values progressively from 2017 to 2020. These changes were informed by reviews of the literature undertaken by all the authors, after which two authors (J.T., T.D.) who were blinded to each other’s assessments assessed the standard of evidence for the interventions implemented. Then, agreement between the two blinded assessments was completed. We also categorised the type of evidence that was used to estimate the effectiveness of interventions, and distinguish between the directly applicable programmatic evidence and the evidence from clinical studies that was extrapolated to the assumed programmatic effect.

### 2.7. Economic Analysis

To estimate the costs of interventions, we used a logistic function to describe the association between cost and coverage of an intervention, as previously described [7]. Estimations of economic inputs were conducted in collaboration with the Fiji NTP and are described in the Appendix A, with the economic parameters presented in Appendix A and the cost-coverage curves presented in Appendix A. The economic analysis was performed from a health care provider perspective, with all costs in 2017 US dollars.

## 3. Results

### 3.1. Calibration and Validation

Figure 1 presents the results of the automatic calibration. The model predictions for TB prevalence remarkably matched the official statistics, although these were not included as calibration targets. Similarly, the reported notifications were also very closely matched by calibration to incidence only, with both the absolute numbers of notified cases as well as the variation in numbers with time accurately reflected by the model. Although the pattern of mortality over time is similar to official statistics, it must be stressed that this model output primarily reflects deaths occurring in the health system, as discussed in the Methods and Discussion sections.

The baseline conditions that were carried forward—which included maintenance of the recent improvements in the case detection rate (above 60%) and the treatment success rates (above 85%)—are expected to result in a slow but steady decline in disease burden. However, this decline is far from that required to meet the End TB Strategy milestones or targets.

### 3.2. Intervention Effectiveness

Figure 2, Table 2 and the Appendix A present the anticipated effectiveness of each of the interventions. The outputs should be interpreted in the context of the differing levels and applicability of evidence for each one, and the caveat that changes in mortality predominantly reflect health system-related deaths. None of the six interventions was sufficient to achieve any but the first set of the End TB Strategy milestones (for 2020). The impact on disease incidence would be particularly limited, as none of the individual interventions could achieve more than an incidence reduction of 3% by 2035 (Table 2), and only a 10% reduction would be achieved by the very optimistic and arguably unrealistic mix of implementing all the programs simultaneously (Scenario 7). In contrast, the effect on disease prevalence was predicted to be more pronounced, as a 17% reduction by 2035 could be achieved by using decentralisation or GeneXpert replacing smear-microscopy (scenarios 2 and 3). Identifying additional cases would increase the number of notifications under all of the scenarios except for treatment support (Scenario 1).

The mortality predictions that are presented in Figure 2 pertain to the mortality rates captured by surveillance systems, such that most of the control interventions increased the observed TB-related mortality by increasing the proportion of active cases known to the health system. The intervention impact on the true mortality (including undetected TB deaths) is presented in Appendix A, with treatment support being the most effective intervention for reducing TB-related deaths. We estimate that the true mortality could be reduced by 23% by 2035 with this program, while none of the other interventions reduced mortality by more than 12%.

The costs of each program are markedly different, ranging from isoniazid preventive therapy (IPT), which is predicted to be very cheap, to GeneXpert-based ACF, which would cost approximately USD 31 million a year.

### 3.3. Impact of T2DM Burden on TB Epidemic

Simulations of changes in the future burden of T2DM predict a marked impact on the burden of TB (Figure 3). The population prevalence of T2DM was changed from the baseline value of 15.6% in 2017 to reach each counterfactual prevalence by 2035, using a sigmoidal function. These changes are associated with considerable changes in disease burden as compared to the baseline prediction for 2035, ranging from an incidence decrease of 16% with T2DM prevalence of 5%, to an incidence increase of 58% when T2DM prevalence reaches 50%.

## 4. Discussion

We predict that TB rates are likely to fall in the coming years in Fiji, but to an extent far from that required to achieve the ambitious goals set by the Fiji NTP. Support for patients under treatment (Scenario 1) is likely to have the greatest impact on observed mortality rates, while raising awareness (Scenario 6) should have the greatest impact on incidence. However, realistic changes in diabetes prevalence are predicted to result in changes to disease burden that are comparable to or greater than these intervention-related effects.

Calibration of our model reveals important insights into TB transmission dynamics in Fiji. Calibration to incidence did not closely match the marked swings in incidence reported in WHO estimates over recent years. However, although prevalence, notifications, and mortality were not calibration targets, our results closely matched each of these important outputs, validating the model results and suggesting that the reported wide swings in incidence may be artefactual. Changes in a range of other programmatic and epidemiological processes over recent years result in a steady decline in each of these markers of TB burden. The recent peak in notifications reflects the improvement in case detection, which—if maintained—would be expected to be followed by a decline in notifications through decreased TB transmission. Moreover, it should be noted that our approach to determining incidence and notifications through the fitting of a mechanistic model would typically result in directly opposite results to the WHO approach of inferring incidence partly from notifications.

Baseline projections of continuing the current programmatic response in Fiji predicts a gradual decline in disease burden that would fall far short of the country’s targets for incidence and mortality for 2020 or the End TB Targets for 2035. Even under the scenario in which all the interventions are simultaneously implemented with high efficacy and coverage (Scenario 7), the End TB Targets still cannot be reached. This is because the pool of latent TB infection (LTBI) acquired before 2016 will continue to drive incidence rates long after this time, especially in the adult population. We estimate that incidence will remain at least 20 per 100,000 in 2035 even if transmission were immediately and completely curtailed. Therefore, any “game changer” intervention must address the issue of reactivation of LTBI, whether by treatment of LTBI or novel vaccination to prevent reactivation.

Increasing the use of IPT to include older children aged between 5–15 years and increasing the coverage in younger children aged between 0–5 years has a substantial impact on reducing incidence, mortality, and prevalence. This is in part attributable to children under 15 years accounting for a relatively large proportion (30%) of the total population [18]. In our economic analysis, this intervention is also cheap, as it requires nothing more than providing isoniazid to patients who are already receiving DOTS visits.

Achieving targets for mortality is problematic, as it is believed that a large proportion of TB-related deaths are unknown to the NTP. Depending on the registry, missing deaths can occur. Using the vital registry, a death can be missed if the NTP does not make the necessary communications with the vital registry and a cause of death other than TB is given. Similarly, missing deaths can occur when a case of TB is not notified to the NTP, only diagnosed post-mortem, and not communicated with the NTP. The proportion of cases that are potentially missed can be estimated using a capture–recapture method, which has been performed in Fiji, suggesting that the mortality rates may be more than three times higher than those estimated by NTP. Indeed, our estimates for total mortality are considerably higher than the estimates for known mortality.

Fiji comprises more than 100 islands, many of which have poor infrastructure and access to care. If hard-to-reach population groups—such as people living on remote islands—are targeted, the strategy would be able to reach larger numbers of missed cases and have a greater impact on TB epidemiology. The targeting of hard-to-reach individuals was considered in two of the scenarios. Scenarios 2 and 5 examined the impact of decentralisation and ACF using mobile health clinics (vans or ferries) equipped with GeneXpert, respectively. The aim of these simulated interventions is to have treatment and diagnostic clinics closer to remote communities. We found that these programs are impactful in improving the detection of cases that would have been otherwise missed, and reducing TB incidence and prevalence. However, these interventions are resource-intensive, and decentralisation is supported by evidence that is of poorer quality or has less direct applicability than many of the other forms.

Replacing sputum smear microscopy with GeneXpert as the primary diagnostic test for passive case detection across the health service (Scenario 3) was found to have a modest effect. This is because there is very little rifampicin resistance, and because cultures are currently being widely used in addition to sputum smear microscopy as a standard diagnostic. Therefore, GeneXpert does not offer greater sensitivity than the current algorithm, and its only effect is to achieve slightly quicker diagnosis, which does not result in a significant epidemiological impact.

Changes in the burden of diabetes could have a marked effect on the burden of TB in Fiji, with a change in prevalence of 10% predicted to lead to a change in TB burden that is equivalent to the most effective intervention simulated. Therefore, targeting diabetes and TB as interwoven co-epidemics in Fiji is critical [19]. 

Limitations of this work include the lack of evidence for the epidemiological and economic parameterisation of some interventions being actively considered by the Fiji NTP. However, the uncertainty analyses performed around the effect of the different interventions did not reveal any changes to the main conclusions. Other limitations are linked to the structural aspects of the model. For example, an explicit spatial structure would be desirable to fully capture the effects of outreach interventions, but this would require additional data that are currently unavailable. Another example is the assumption of homogeneous mixing, which may not be realistic, as there has been evidence of age-assortative mixing in other settings [20]. Again, the lack of local data on social mixing at the time of the model elaboration was the main barrier to heterogeneous mixing implementation. Lastly, as transmission rates continue to fall and case numbers per year decline to double digits, stochastic simulations will be important to capture the full range of possible outcomes. Some of the findings presented here may be applicable to other parts of the Pacific, due to the high rates of diabetes and the geographical challenges for case detection that are common to many such countries [21]. However, there are also dramatic differences between Pacific countries, including huge differences in disease rates between the highest (e.g., Kiribati, the Marshall islands) and lowest (e.g., Samoa) burden countries [22]. 

## 5. Conclusions

If current programmatic responses are continued, we predict further gradual declines in the TB burden in Fiji, although these will be well short of the SDGs and End TB targets. Finding additional TB cases over and above the current (mostly passive) case detection is an important component of reducing TB burden. However, addressing the underlying burden of LTBI will be essential if the country is to achieve its disease-related goals. The TB-diabetes co-epidemic is also of particular importance in this setting, and could derail attempts to achieve the ambitious targets of the Fiji NTP.

## Figures and Tables

**Figure 1 tropicalmed-04-00071-f001:**
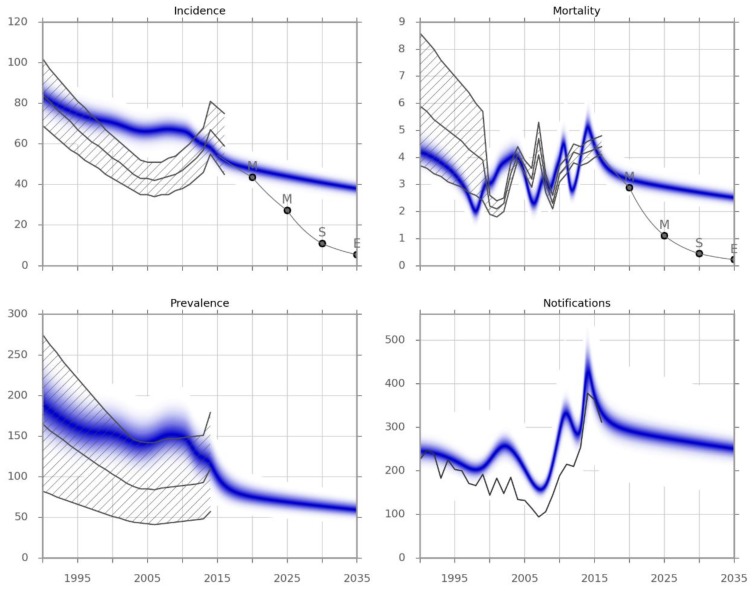
Model calibration results. Incidence and observed mortality (per 100,000 per year), prevalence (per 100,000) and the number of notifications by calendar year. The blue-shaded areas represent the calibrated model predictions obtained from the Metropolis simulation. The grey lines represent point estimates, and the hatched areas represent the confidence limits for each indicator from the Global TB Report 2017. The light grey line indicates the epidemic trajectory that would be required to achieve the different targets, and is a piecewise exponential function. M, Milestone; S, Sustainable Development Goal; E, End TB Strategy Target.

**Figure 2 tropicalmed-04-00071-f002:**
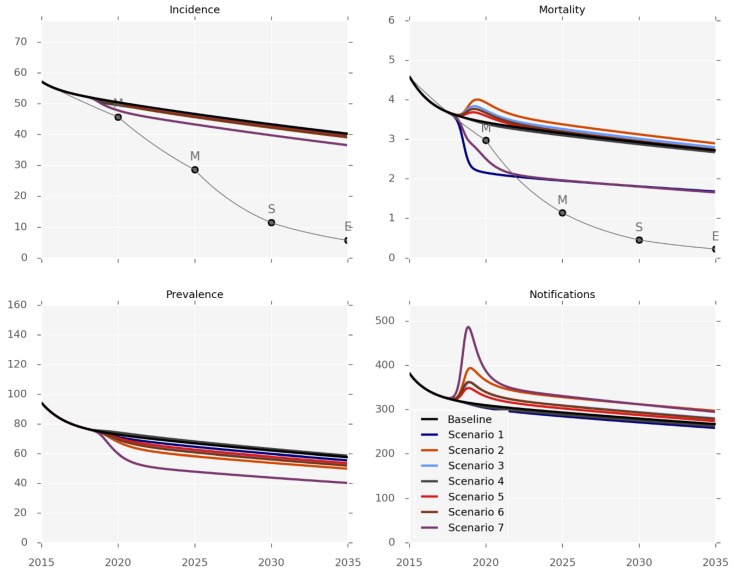
Intervention effectiveness. Incidence and observed mortality (per 100,000 per year), prevalence (per 100,000) and number of notifications by calendar year.

**Figure 3 tropicalmed-04-00071-f003:**
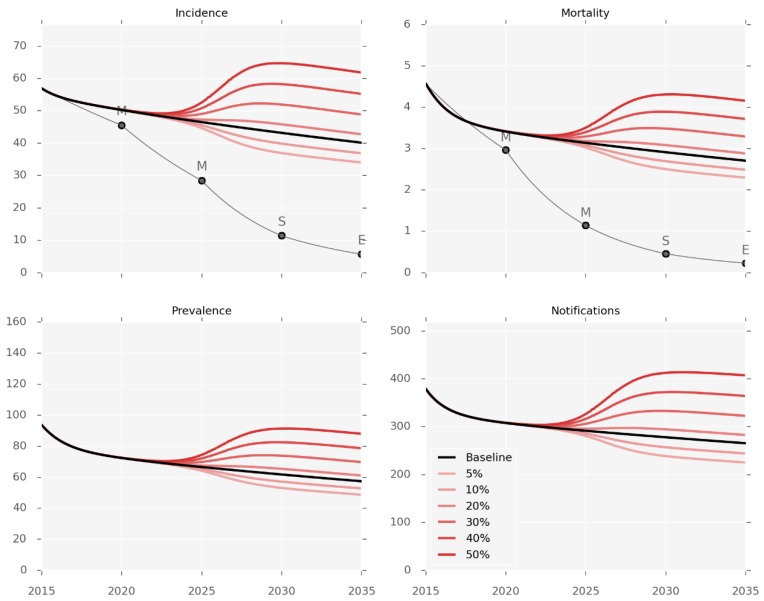
Diabetes prevalence counterfactuals. Incidence and observed mortality (per 100,000 per year), prevalence (per 100,000), and number of notifications by calendar year. Red lines represent the different levels of T2DM prevalence varying from 5% to 50%. The baseline scenario (15.6% T2DM prevalence) is shown in black.

**Table 1 tropicalmed-04-00071-t001:** Intervention implementation.

Scenario	Description	Level of Evidence *	Applicability of Evidence	Primary Evidence Source	Coverage Achievable (by 2020)	Model Implementation
1. Support for patients under treatment	Health worker visits patients upon their return home from the hospital, treatment adherence checks, regular clinic appointments	2	Programmatic	Thiam et al. 2007 [12]	100%	Decrease in all treatment outcomes other than success (i.e., death and non-death unfavourable) by 43% (range 21% to 89%)
2. Decentralisation of care	Transfer of diagnostic and treatment facilities to most remote communities to remove access barriers	5	Mechanism-based reasoning	N/A	70%	Increase in case detection rate from baseline value to idealised value informed by those reported by the best-performing regional TB programs, increasing to 75% (range 65 to 90%)
3. GeneXpert replaces smear	GeneXpert replaces smear microscopy as the primary diagnostic test for passive case detection across the health service	1	Programmatic	Boehme et al. 2011 [13]	100%	Decrease in smear-negative cases missed by diagnostic algorithm by 76.9% (range 56.3% to 100%), decrease in time to treatment commencement to seven days (range one to 30 days)
4. Isoniazid preventive therapy (IPT)	Expansion of coverage of IPT for contacts of pulmonary cases from existing coverage levels (23.6% of under 5 years old in 2014) to broad coverage of all cases under 15 years old	1	Clinical	Sollai et al. 2014, Smieja et al. 2000 [14,15]	80%	Proportion of infection occurring in households is 60% (range 40% to 80%), sensitivity of testing for LTBI 70% (range 70% to 80%), efficacy of treatment is 60% (range 48% to 69%)
5. Active case finding (ACF)	Van and ferry-based outreach to detect previously unrecognised cases	2	Programmatic	Corbett et al. 2010 [16]	50%	GeneXpert test performed in individuals presenting any TB-related symptom (27% of individuals) Proportion of cases diagnosed during a single ACF round is 23% (range 10% to 30%), with one round of ACF conducted per year
6. Awareness raising	Broad mass media campaign and community engagement to improve community knowledge, attitudes, and practices associated with TB	3	Programmatic	Jaramillo et al. [17]	50%	Rate of presentation for care for undiagnosed increases 1.52-fold (range 1.34 to 1.92-fold) the baseline value
7. Combination of scenarios 1–6	All the interventions described above were implemented simultaneously					

* Level of evidence was graded according to the Oxford Centre for Evidence-Based Medicine 2011 Levels of Evidence. Level 1 indicates the strongest evidence, whereas level 5 indicates the weakest evidence. Abbreviations: ACF, active case finding; IPT, isoniazid preventive therapy; LTBI, latent tuberculosis infection; N/A, not applicable; TB: tuberculosis.

**Table 2 tropicalmed-04-00071-t002:** Predicted intervention effectiveness. Values in brackets represent 95% simulation intervals.

Intervention *	Incidence in 2035		True mortality in 2035		Observed Mortality in 2035		Additional Intervention Costs
	Per 100,000 per Year	Relative Change (%)	Per 100,000 per Year	Relative Change (%)	Per 100,000 per Year	Relative Change (%)	(USD, per Year)
Baseline projection	39.1	-	7.1	-	2.6	-	-
1 (Treatment support)	38.4 (38.2 to 38.7)	−1.7 (−2.3 to −1.0)	5.5 (5.0 to 6.2)	−22.5 (−29.7 to −12.4)	1.2 (0.8 to 1.9)	−53.2 (−70.2 to −29.3)	441,096
2 (Decentralisation)	38.4 (36.1 to 38.9)	−1.8 (−7.6 to −0.6)	6.6 (5.3 to 6.9)	−7.3 (−25.7 to −3.2)	2.8 (2.7 to 2.8)	6.4 (3.8 to 7.2)	532,825
3 (GeneXpert)	38.3 (38.2 to 38.5)	−2.1 (−2.3 to −1.5)	6.8 (6.7 to 6.9)	−5.0 (−5.6 to −3.7)	2.7 (2.7 to 2.7)	2.9 (2.2 to 3.2)	2,046,850
4 (IPT)	38.3 (38.2 to 38.5)	−2.0 (−2.1 to −1.9)	6.8 (6.7 to 6.9)	−4.8 (−5.0 to −4.7)	2.7 (2.7 to 2.7)	2.8 (2.7 to 2.9)	24,688
5 (ACF)	38.3 (37.9 to 38.6)	−2.2 (−3.1 to −1.3)	6.7 (6.5 to 6.9)	−6.5 (−9.0 to −3.7)	2.6 (2.6 to 2.6)	−0.1 (−0.2 to −0.1)	31,414,371
6 (Awareness)	37.7 (37.3 to 38.3)	−3.5 (−4.6 to −2.2)	6.3 (6.1 to 6.6)	−11.5 (−15.1 to −7.2)	2.7 (2.6 to 2.7)	0.7 (0.4 to 0.9)	4,576,185**

* Refer to Table 1 for descriptions of the scenarios. ** Estimate of costs is highly uncertain. Abbreviations: ACF, active case finding; IPT, isoniazid preventive therapy.

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
