# Peer review of "Strategic Planning for Tuberculosis Control in the Republic of Fiji"

_tropicalmed, 2019, doi:10.3390/tropicalmed4020071_

Round 1
Reviewer 1 Report
Ragonnet et al
Main comments
- I found a good level of detail of the AuTuMN software in the main text and supplementary. However, it would also be good to see how the model fits with the TB MAC benchmarking for country level assessments: http://tb-mac.org/tb-mac-resource/guidance-for-country-level-tb-modelling/. Can this be added to the start of the supplementary at least with some comments on achieving the targets in the main text?
- Can Fiji really be modelled without spatial structure?
- Does T2DM change in your baseline model? I think not – but surely it would have been good to combine and explore interventions and T2DM changes?
- Can you add more quantification? Instead of similar magnitude or large magnitude can you put in % change values?
Minor comments
- The ordering of the references to the supplementary were odd (e.g. Figure S3 and then jumping to Figure S9?)
- What parameters were split by age?
- Was paediatric < 15 years?
- Can you add a sentence describing what a Metropolis algorithm is into the main text and a reference to the method?
- Where was your TB incidence data from? Did you only calibrate to this? What limitations does this impose on your interpretations?
- How did you assess fit to mortality if they were not in the calibration algorithm? Just visually? Can this really be used rigorously? What does “remarkably match” mean?
- I did not understand the point of the “standard” or “level” of evidence? Was this just for whether an intervention worked or for the ability to parameterise your model to capture this?
- Page 6 was missing from my pdf but I don’t think there was anything on this (Table 1 straight into Results)
- Intervention 3 mentions households – but the model was not structured by household was it?
- Intervention 6: over what period did the rate of presentation increase?
- Interventions 5&6: how often? One round?
- Figure 1: I found this very hard to read. The light grey line was very hard to see on a black and white printout. The big fluctuations in the right hand figures (mortality and notifications) seem to completely miss the underlying dynamics whilst being clearly in all of your simulations. Can you explain this?
- Figure 1: This is not a great fit to the data. Can you explain why the algorithm failed? It appears to overestimate incidence.
- Can you add some quantification into your results? E.g. how far is your model prediction from the required decline? E.g. what is “much greater in magnitude”?
- Figure 2: the colours are hard to see on a b&w printout.
- Table 2: why are there no ranges on the costs?
- Impact of T2DM: can you make it clear that it has the same magnitude of impact but in the opposite direction?
- Where is the sigmoidal function that use for the T2DM increase? Link to the supplementary figure. Did you test other functions?
- Discussion: from Table 2, doesn’t awareness (6) have the biggest impact on incidence at -3.5% instead of decentralisation of care (2) at -1.8%?
- Can you add intervention number into the discussion?
- If the swings in incidence are “artefactual” what does this mean for your model interpretation?
- Can you add a reference to the capture-recapture study in Fiji?
- You discuss that latent carriage should be targeted, and then later IPT. Why isn’t IPT effective for tackling latent carriage? Can you discuss these needs and this intervention together?
- What are the “hard-to-reach” populations (line 109)?
- Line 126 “co-epidemics”
- could you add some more discussion of limitations: e.g. model structure, model fitting to only incidence, parameterisation/implementation of interventions
- Author contributions, funding and acknowledgements were not complete
Author Response
Reviewer 1
Main comments
R1C1. I found a good level of detail of the AuTuMN software in the main text and supplementary. However, it would also be good to see how the model fits with the TB MAC benchmarking for country level assessments: http://tb-mac.org/tb-mac-resource/guidance-for-country-level-tb-modelling/. Can this be added to the start of the supplementary at least with some comments on achieving the targets in the main text?
We agree that benchmarking is an important process and detailed comparison of our model outputs to the TB MAC benchmarks is planned for the future applications of our model. However, the TB MAC benchmarks did not exist at the time that this version of the model was implemented so the system was not designed to produce the related estimates and to use the TB MAC benchmarks for calibration. In our current country applications (Bhutan and Mongolia), we are working to output the benchmarks routinely as a core part of our software platform.
R1C2. Can Fiji really be modelled without spatial structure?
We acknowledge the particular characteristics of the Fijian geography. However, spatial stratification was not identified as a priority feature by the national TB program and its implementation is significantly hampered by the lack of data at a sufficiently high resolution. Specifically, to implement a fully spatially explicit (presumably patch-based compartmental) model, information would be required on population spatial distribution, population mixing, and prevalence of age groups and comorbidities by geographical region. Such information is not available for Fiji. In addition, significant work on coding would be essential to implement these factors as it is not currently a core part of the AuTuMN platform.
Despite these limitations, we did approach the Fiji National TB Program in the months following the application to propose a geospatial analysis with different modelling techniques. Unfortunately, we were not able to obtain sufficient data to complete this geospatial exercise.
In response to this comment, we have now included the absence of spatial structure in the study limitations:
“Other limitations are linked to the structural aspects of the model. For example, explicit spatial structure would be desirable to fully capture the effects of outreach interventions, but this would require additional data that are currently unavailable” (lines 292-294)
R1C3. Does T2DM change in your baseline model? I think not – but surely it would have been good to combine and explore interventions and T2DM changes?
The changes in T2DM are captured in the baseline model. Its time-variant prevalence is presented in Figure S9.
R1C4. Can you add more quantification? Instead of similar magnitude or large magnitude can you put in % change values?
We have added the following sentence to better quantify the impact of diabetes on the TB epidemic:
“These changes are associated with considerable changes in disease burden as compared to the baseline prediction for 2035, ranging from an incidence decrease of 16% with T2DM prevalence of 5%, to an incidence increase of 58% when T2DM prevalence reaches 50%.” (lines 216-219)
Minor comments
R1C5. The ordering of the references to the supplementary were odd (e.g. Figure S3 and then jumping to Figure S9?)
We have now updated the numbering of the figures in the supplement.
R1C6. What parameters were split by age?
The age-specific model implementations are the waning of BCG-related immunity (mentioned in main text at lines 94-95), the diabetes comorbidity that applies to the 25+ years age group only (lines 87-88), age-dependent infectiousness (lines 92-93) and the rates of progression from latent infection to active disease (lines 101-102).
R1C7. Was paediatric < 15 years?
Yes, paediatric TB is defined based on the most commonly used convention (<15 years old). We now specify this in the main text:
“All paediatric (<15 years old) and smear-negative TB patients …” (line 92).
R1C8. Can you add a sentence describing what a Metropolis algorithm is into the main text and a reference to the method?
We have added the following sentences in the main text as well as a reference to a technical description of the algorithm:
“A Metropolis algorithm is a Markov chain Monte Carlo method that allows to sample from the posterior distribution of the model parameters [11]. That is, it identifies the model parameter sets that are best able to replicate trends observed in the data. Details about the Metropolis algorithm used in this study are available in the Supplement (pages 12-14).” (lines 126-129)
R1C9. Where was your TB incidence data from? Did you only calibrate to this? What limitations does this impose on your interpretations? How did you assess fit to mortality if they were not in the calibration algorithm? Just visually? Can this really be used rigorously? What does “remarkably match” mean?
We used WHO estimates of TB incidence to calibrate our model. The following sentence was added to the text:
“We used a Metropolis algorithm to calibrate the model to align with the TB incidence estimated by the WHO for years 2010 to 2016” (lines 122-123).
TB incidence was the only indicator included in the calibration process. The other disease burden indicators (mortality, notifications and prevalence) emerged naturally and were used for validation. We observed that our predictions for these other, non-calibration indicators closely matched official statistics, which we believe is a sign of reliability and robustness of the model.
R1C10. I did not understand the point of the “standard” or “level” of evidence? Was this just for whether an intervention worked or for the ability to parameterise your model to capture this?
Please see the footnote of Table 1: “Level of evidence was graded according to the Oxford Centre for Evidence-Based Medicine 2011 Levels of Evidence. Level 1 indicates strongest evidence, whereas level 5 indicates weakest evidence.”
This means that this is the quality of the evidence that is evaluated and not the effectiveness of the intervention. For example, an intervention could be associated with a very high level of evidence even if entirely ineffective, provided that there is strong evidence demonstrating its ineffectiveness.
R1C11. Page 6 was missing from my pdf but I don’t think there was anything on this (Table 1 straight into Results)
This was indeed a formatting issue and there is no content between Table 1 and the Results section.
R1C12. Intervention 3 mentions households – but the model was not structured by household was it?
The reviewer is right that there is no household structure in the model, although one intervention does apply to household contacts (Intervention 4, IPT). The effectiveness of this intervention is calculated by estimating the proportion of transmission events that occur within households (60%, range 40-80). This means that we account for the fact that not all contacts of TB patients can be traced, even if the model does not explicitly distinguish traceable and untraceable contacts.
R1C13. Intervention 6: over what period did the rate of presentation increase?
We assumed that the rate of presentation increases over the period 2017-2020. We have added the following to the text in order to be more specific about the scale-up period:
“The interventions were modelled by changing relevant model parameters from their baseline values, using continuous scale-up function increase values progressively from 2017 to 2020” (lines 141-143)
R1C14. Interventions 5&6: how often? One round?
We thank the reviewer for noting that we did not provide this information. We have now added the missing details in Table 2:
“One round of ACF is conducted per year.”
R1C15. Figure 1: I found this very hard to read. The light grey line was very hard to see on a black and white printout. The big fluctuations in the right hand figures (mortality and notifications) seem to completely miss the underlying dynamics whilst being clearly in all of your simulations. Can you explain this? This is not a great fit to the data. Can you explain why the algorithm failed? It appears to overestimate incidence.
We have produced a new Figure 1 to address the issues raised by the reviewer.
Although the country stakeholders, including the members of the Fiji National TB Program authoring the current manuscript, were convinced that the latest WHO estimates of TB incidence were appropriate, they were not confident that the U-shaped curve observed in the incidence reported by WHO between 1995 and 2015 reflected the recent epidemiology of TB in the country. We therefore prioritised fitting to the estimates of the most recent years in our calibration algorithm, considering only the period 2010-2016 for calibration (main text, lines 122-123).
R1C16. Can you add some quantification into your results? E.g. how far is your model prediction from the required decline? E.g. what is “much greater in magnitude”?
The following text was added in the Results section:
“These changes are associated with considerable changes in disease burden as compared to the baseline prediction for 2035, ranging from an incidence decrease of 16% with T2DM prevalence of 5%, to an incidence increase of 58% when T2DM prevalence reaches 50%.” (lines 216-219)
R1C17. Figure 2: the colours are hard to see on a b&w printout.
Figure 2 has been updated to improve readability.
R1C18. Table 2: why are there no ranges on the costs?
Our model does not account for uncertainty in the input costs. This means that the cost variability that we could possibly report would result from the epidemiological uncertainty only, without accounting for the actual economic uncertainty. To avoid providing any misleading information, we chose not to report any variations in the cost outputs. In subsequent AuTuMN analyses (e.g. Bulgaria) we have presented economic uncertainty. However, this functionality was not available at the time of the Fiji application and so we did not estimate uncertainty intervals around our cost inputs for this application.
R1C19. Impact of T2DM: can you make it clear that it has the same magnitude of impact but in the opposite direction?
The text added in response to comment R1C4 should address this issue.
R1C20. Where is the sigmoidal function that use for the T2DM increase? Link to the supplementary figure. Did you test other functions?
We have added a figure to present the scale-up curves used to simulate the different levels of T2DM prevalence. See Supplement, Figure S13.
R1C21. Discussion: from Table 2, doesn’t awareness (6) have the biggest impact on incidence at -3.5% instead of decentralisation of care (2) at -1.8%?
We thank the reviewer for identifying this error. The text has now been changed in the Discussion.
“Support for patients under treatment (Scenario 1) is likely to have the greatest impact on observed mortality rates, while raising awareness (Scenario 6) should have the greatest impact on incidence” (lines 228-230)
R1C22. Can you add intervention number into the discussion?
We have added the scenario numbers in the discussion section.
R1C23. If the swings in incidence are “artefactual” what does this mean for your model interpretation?
As mentioned in response to comment R1C15, we did not aim to capture the U-shaped curve observed in the WHO-estimated incidence when calibrating the model. This implies that our model estimates would not be affected if this U-shape was artefactual.
R1C24. Can you add a reference to the capture-recapture study in Fiji?
The reference was added in the introduction section (line 37).
R1C25. You discuss that latent carriage should be targeted, and then later IPT. Why isn’t IPT effective for tackling latent carriage? Can you discuss these needs and this intervention together?
We acknowledge that the ordering of the paragraphs did not allow the reader to make the link between our discussion of latent infection and that of IPT effect easily. We have re-ordered the section such that IPT is now discussed just after the issue of latent carriage.
The reason why IPT is not able to tackle latent carriage is that “the pool of latent TB infection (LTBI) acquired before 2016 will continue to drive incidence rates long after this time” (lines 248-249). This explanation was already given in the previous version of the manuscript but we acknowledge that it could be improved by specifying that this issue would mostly affect the adult population. This addition will help to make the link with the following section that discusses the effect of IPT provided to <15 years old individuals. The sentence has now become:
“This is because the pool of latent TB infection (LTBI) acquired before 2016 will continue to drive incidence rates long after this time, especially in the adult population.”
R1C26. What are the “hard-to-reach” populations (line 109)?
We have now included a more specific description of the “hard-to-reach” populations:
“Fiji is composed of more than 100 islands, many of which have poor infrastructure and access to care. If hard-to-reach population groups, such as people living on remote islands, are targeted, it would be able to reach larger numbers of missed cases and have a greater impact on TB epidemiology.” (lines 268-271)
R1C27. Line 126 “co-epidemics”
This has been changed.
R1C28. could you add some more discussion of limitations: e.g. model structure, model fitting to only incidence, parameterisation/implementation of interventions
We have added more discussion of the study limitations:
“Limitations of this work include the lack of evidence for epidemiological and economic parameterisation of some interventions being actively considered by the Fiji NTP. However, the uncertainty analyses performed around the effect of the different interventions did not reveal any changes to the main conclusions. Other limitations are linked to the structural aspects of the model. For example, explicit spatial structure would be desirable to fully capture the effects of outreach interventions but this would require additional data that are not available for the moment. Another example is the assumption of homogeneous mixing which may not be realistic, as there has been evidence of age-assortative mixing in other settings [20]. Again, the lack of local data on social mixing at the time of the model elaboration was the main barrier to heterogeneous mixing implementation. Last, as transmission rates continue to fall and case numbers per year decline to double digits, stochastic simulations will be important to capture the full range of possible outcomes. Some of the findings presented here may be applicable to other parts of the Pacific, due to the high rates of diabetes and geographical challenges for case detection common to many such countries [21]. However, there are also dramatic differences between Pacific countries, including marked differences in disease rates between the highest (e.g. Kiribati, the Marshall islands) and lowest (e.g. Samoa) burden countries [22].”
R1C29. Author contributions, funding and acknowledgements were not complete
These sections have now been completed.
Reviewer 2 Report
The manuscript evaluates the impact and costs of different control interventions towards the achievement of international targets (Sustainable Development Goals, EndTB Strategy) for tuberculosis [TB] incidence reduction in the Republic of Fiji. The authors used a previously published mathematical model of TB transmission dynamics taking into account a large number of factors representing the complex epidemiology of TB, including demographic dynamics and the staggering prevalence of DM in the Fiji. The paper is, for the most part, clearly written and concludes that current and prospective control interventions will fall short of the target objectives, mainly because of the high numbers of TB cases that are expected to reactivate from latently infected individuals.
Despite the impressive amount of detail that is included in the model, which I commend, I believe there are a number of shortcomings that would deserve to be better addressed. Overall, I recommend a revision of the manuscript taking into account the main points below, either with new computations or with convincing discussions/demonstrations that the criticisms only has a minor effect on model estimates.
1) A main issue is the lack of a proper explanation on how the free parameters of the model were calibrated. The authors state that they "accept model runs based on a Metropolis Hastings algorithm" without specifying how they define the likelihood, how the priors for free parameters (distributions and metaparameters, fig S11) were chosen and what are the posterior distributions of calibrated parameters. Furthermore, posterior distributions of parameters are not actually used for projecting the uncertainty in the impact of each scenario, which is the main advantage of using Bayesian parameter estimation (the authors use projections from maximum likelihood parameters instead);
1.1) As a minor side note, I can't understand why the authors included the starting point of the simulation as a free model parameter: in practice, because all parameters are fixed until at least 1928 (when BCG vaccination starts), the authors should be able to bring the model to epidemiological equilibrium by running it for a time of about twice the life expectancy (i.e. starting in 1800 should be sufficient to get rid of this parameter);
2) For what concerns structural aspects of the model, I applaud the authors for including demographic dynamics over time (too many TB models are still neglecting this issue, which is critical for chronic infections); however, I am concerned with the use of mortality rates that are equal for all age groups, which seems quite inaccurate, leading to a negative exponential age-distribution of the population. It would be useful to verify the match between model-predicted and observed population distributions by age groups, possibly at multiple time points. Considering the importance of age in TB epidemiology (and in the considered model), it is very important to capture correctly the relative weights of each age group to the overall incidence.
3) On a similar topic, I haven't seen references to heterogeneous mixing by age. Air-borne infections are highly assortative by age especially during childhood and young adulthood; using homogeneous mixing is known to overestimate transmission, so that incidence can be fitted with a lower contact rate compared to heterogeneous mixing. This might be a major source of bias for the evaluation of control interventions. I recommend checking data provided in Prem K, Cook AR, Jit M. Projecting social contact matrices in 152 countries using contact surveys and demographic data. PLoS computational biology, 2017 for estimates of contact matrices that may be relevant for the Fiji.
4) Again on age-specific assumptions, but more related to the natural history of TB: the authors use age-independent proportions of smear positive/negative and extrapulmonary TB and then assume arbitrary scaling factors for the infectiousness of children (<5yrs and 5-15 yrs); as far as I know, children are less infectious exactly because they mostly develop extrapulmonary and smear-negative TB. Therefore the relative infectiousness should emerge as a result of different compositions of disease manifestations rather than from ad-hoc parameters; data on the proportions of smear-negative and extrapulmonary TB in children should be easily available.
5) A number of additional model estimates would be useful for the interpretation of results (both to evaluate the robustness of the model and to have insights on the TB epidemiology); in particular:
* what is the distribution of incident cases and TB deaths by age (at selected time points)?
* what is the proportion of reactivated vs. active TB cases over time?
* how does the proportion of relapse/reinfections increase as a higher number of individuals are treated in different scenarios?
* what is the reduction in the estimated total mortality, rather than (or in addition to) the observed one?
* what is the proportion of total averted TB deaths by age in different scenarios?
* costing information should be given not only in absolute terms, but also relative to the number of averted cases, averted deaths and life-years saved (possibly disability-adjusted); the latter in particular would be useful to weigh more the value of interventions that are more effective in saving infants and young children compared to elderly people;
* there should be an evaluation of the variability in the expected outcome (e.g. average + 95%CI + maximum likelihood) for different strategies, to evaluate how parameter uncertainty may impact on the estimated differences.
6) In Figure 3, the timing of TD2M changes in prevalence seems excessively sharp; according to data presented in the Supplementary Materials, the prevalence of type 2 diabetes changed from 8% to 16% between 1980 and 2000 and remained quite stable in the last 15; if I'm not mistaken, considered scenarios envision a 10 to 35% increase in prevalence within the next 5 years?
Minor comments:
- It is not clear to me whether the adopted model was taken "as is" from the published version or if adjustments in structure and/or parameter choices were done compared to other works; in the latter case, differences between previously peer-reviewed modeling choices and this work should be highlighted to allow a fairer review process;
- In Figure 2 in the main text, the color legend should be displayed graphically rather than textually to facilitate the association of the curves to different scenarios. In addition, the color palette makes many scenarios hardly distinguishable (e.g. 2, 5 and 6), also considering that the curves are very close to each other.
Author Response
Opening comment. The manuscript evaluates the impact and costs of different control interventions towards the achievement of international targets (Sustainable Development Goals, EndTB Strategy) for tuberculosis [TB] incidence reduction in the Republic of Fiji. The authors used a previously published mathematical model of TB transmission dynamics taking into account a large number of factors representing the complex epidemiology of TB, including demographic dynamics and the staggering prevalence of DM in the Fiji. The paper is, for the most part, clearly written and concludes that current and prospective control interventions will fall short of the target objectives, mainly because of the high numbers of TB cases that are expected to reactivate from latently infected individuals.
Despite the impressive amount of detail that is included in the model, which I commend, I believe there are a number of shortcomings that would deserve to be better addressed. Overall, I recommend a revision of the manuscript taking into account the main points below, either with new computations or with convincing discussions/demonstrations that the criticisms only has a minor effect on model estimates.
1) A main issue is the lack of a proper explanation on how the free parameters of the model were calibrated. The authors state that they "accept model runs based on a Metropolis Hastings algorithm" without specifying how they define the likelihood, how the priors for free parameters (distributions and metaparameters, fig S11) were chosen and what are the posterior distributions of calibrated parameters. Furthermore, posterior distributions of parameters are not actually used for projecting the uncertainty in the impact of each scenario, which is the main advantage of using Bayesian parameter estimation (the authors use projections from maximum likelihood parameters instead);
We thank the reviewer for pointing out the lack of details provided about the Metropolis Hastings algorithm. We have now extended the section presenting the algorithm in the Supplement. Namely, the prior distributions are now clearly introduced, the likelihood function is now defined and the posterior estimates of the parameters are now presented (Figure S12).
The interpretation of the reviewer is correct about the fact that the posterior distributions of the parameters were not used to evaluate the effect of the interventions. Indeed, our aim was to evaluate the impact of the uncertainty in intervention efficacy separately from the epidemiological uncertainty considered in the Metropolis Hastings algorithm as we believe that this makes the reported messages easier to understand.
1.1) As a minor side note, I can't understand why the authors included the starting point of the simulation as a free model parameter: in practice, because all parameters are fixed until at least 1928 (when BCG vaccination starts), the authors should be able to bring the model to epidemiological equilibrium by running it for a time of about twice the life expectancy (i.e. starting in 1800 should be sufficient to get rid of this parameter);
We believe that simulating the Fijian TB epidemic as an equilibrium before 1928 would not be accurate, given the Fijian immigration history involving colonisation and the arrival of tens of thousands of Indian migrants in the beginning of the last century. Such waves of migration are very likely to have contributed to important changes in the historical trends of the TB epidemic. We therefore believe that it is appropriate to incorporate model starting point as an uncertain parameter.
2) For what concerns structural aspects of the model, I applaud the authors for including demographic dynamics over time (too many TB models are still neglecting this issue, which is critical for chronic infections); however, I am concerned with the use of mortality rates that are equal for all age groups, which seems quite inaccurate, leading to a negative exponential age-distribution of the population. It would be useful to verify the match between model-predicted and observed population distributions by age groups, possibly at multiple time points. Considering the importance of age in TB epidemiology (and in the considered model), it is very important to capture correctly the relative weights of each age group to the overall incidence.
We acknowledge that the negative exponential age-distribution is not realistic and we will use age-specific mortality rates in our future model implementations. However, for Fiji, this assumption led to a demographic profile very similar to the observed demographic profile.
Following the reviewers’ suggestion, we have generated the age-distribution of the simulated population over time (Figure S14) and we have included a figure to compare the age-distribution of the Fiji 2017 population predicted by the model with that observed from census data (Figure S15). This illustrates how closely matched the simulated age group proportions are to the observed proportions. There are clear reasons for this. Fiji is still transitioning in its demographics from high birth/high death rate to high birth/low death rate. These were captured by our model (even without age-specific death rates) and are the main drivers of demography currently. Finally, the use of a high birth and exponential death rate mostly impacts the shape of the middle ages/older ages, and our model was rather course, grouping all 25+ into one age bracket. Hence, we believe that for Fiji, our exponential death model was adequate.
We acknowledge this is a limitation and we plan to introduce age-specific death rates in subsequent applications of the model, where it will be deployed in countries with different demographic profiles.
3) On a similar topic, I haven't seen references to heterogeneous mixing by age. Air-borne infections are highly assortative by age especially during childhood and young adulthood; using homogeneous mixing is known to overestimate transmission, so that incidence can be fitted with a lower contact rate compared to heterogeneous mixing. This might be a major source of bias for the evaluation of control interventions. I recommend checking data provided in Prem K, Cook AR, Jit M. Projecting social contact matrices in 152 countries using contact surveys and demographic data. PLoS computational biology, 2017 for estimates of contact matrices that may be relevant for the Fiji.
We recognise the importance of heterogeneous mixing in transmission dynamic models but unfortunately, there was no data available on social mixing for Fiji when we elaborated the model for Fiji. We are aware of the country-specific estimates produced by Prem and colleagues, as we currently use them in another project. However, it is to be noted that these estimates were based on data from eight Western Europe countries and that contact rates for all other countries were obtained by extrapolation. We did not use these estimates for Fiji because of the dramatically different socio-economic profile of the country as compared to the eight countries in which the contact surveys were undertaken.
This limitation is now clearly stated in the discussion section:
“Another example is the assumption of homogeneous mixing which may not be realistic, as there has been evidence of age-assortative mixing in other settings [Ref]. Again, the lack of local data on social mixing at the time of the model elaboration was the main barrier to heterogeneous mixing implementation.” (lines 294-297).
4) Again on age-specific assumptions, but more related to the natural history of TB: the authors use age-independent proportions of smear positive/negative and extrapulmonary TB and then assume arbitrary scaling factors for the infectiousness of children (<5yrs and 5-15 yrs); as far as I know, children are less infectious exactly because they mostly develop extrapulmonary and smear-negative TB. Therefore the relative infectiousness should emerge as a result of different compositions of disease manifestations rather than from ad-hoc parameters; data on the proportions of smear-negative and extrapulmonary TB in children should be easily available.
We agree that children are usually smear negative, partly because of the nature of their tuberculosis and partly because they reflexively swallow sputum and cannot produce sputum smear. Either way they are less infectious. From the model dynamics point of view, the way we have coded this will have the same properties as changing all pulmonary cases to smear negative and then further reducing the infectiousness of smear negative cases. We have now reworded this section in the supplementary material (Age differences/infectiousness) to ensure it reflects the reality of the situation as follows:
For these reasons, we reduce the infectiousness of children by one order of magnitude (i.e. 0.1 times that of adults) and apply this to simulate the under-5 age group. Although we maintain the division into smear negative/smear positive and extra-pulmonary in this age group to reflect the same qualitative status as in the adult age group, we acknowledge all of this group is likely to be smear negative most of the time, and use the (0.1 fold) multiplier to reflect this.
5) A number of additional model estimates would be useful for the interpretation of results (both to evaluate the robustness of the model and to have insights on the TB epidemiology); in particular:
* what is the distribution of incident cases and TB deaths by age (at selected time points)?
* what is the proportion of reactivated vs. active TB cases over time?
* how does the proportion of relapse/reinfections increase as a higher number of individuals are treated in different scenarios?
* what is the reduction in the estimated total mortality, rather than (or in addition to) the observed one?
* what is the proportion of total averted TB deaths by age in different scenarios?
* costing information should be given not only in absolute terms, but also relative to the number of averted cases, averted deaths and life-years saved (possibly disability-adjusted); the latter in particular would be useful to weigh more the value of interventions that are more effective in saving infants and young children compared to elderly people;
* there should be an evaluation of the variability in the expected outcome (e.g. average + 95%CI + maximum likelihood) for different strategies, to evaluate how parameter uncertainty may impact on the estimated differences.
We have produced additional figures in order to increase the interpretability of our results, as suggested by the reviewer. These figures include:
· the distribution of incident cases by age over time (Figure S16)
· the distribution of TB notifications by age over time (Figure S17)
· the distribution of TB deaths by age over time (Figure S18)
· the proportion of TB incidence that is due to fast progression versus late reactivation over time (Figure S19)
· the scale-up profiles used to simulate the different scenarios of diabetes prevalence (Figure S13)
· the posterior distributions of the calibrated parameters (Figure S12)
· the age-distribution of the simulated population over time (Figure S14)
· a comparison between the age-distribution predicted by the model and that obtained from census data (Figure S15)
· the reduction in the estimated total mortality in addition to the observed one was already presented in Table 2.
6) In Figure 3, the timing of TD2M changes in prevalence seems excessively sharp; according to data presented in the Supplementary Materials, the prevalence of type 2 diabetes changed from 8% to 16% between 1980 and 2000 and remained quite stable in the last 15; if I'm not mistaken, considered scenarios envision a 10 to 35% increase in prevalence within the next 5 years?
We acknowledge that the changes in diabetes prevalence were too sharp and we have now changed the scale-up functions used to generate the associated scenarios. Figure S13 presents these scale-up functions which now reach the new levels of TD2M prevalence in 2035.
Minor comments:
- It is not clear to me whether the adopted model was taken "as is" from the published version or if adjustments in structure and/or parameter choices were done compared to other works; in the latter case, differences between previously peer-reviewed modeling choices and this work should be highlighted to allow a fairer review process;
The model is a particular configuration of the software described in Trauer et al. 2017. Although many parameters and model elaborations are common to all countries simulated with the AuTuMN platform, the model is adapted to each setting by selecting the optional features to be implemented (e.g. drug resistance, age-stratification, diabetes …) and by using setting-specific parameter values.
We believe that the configuration used to simulate TB epidemiology in Fiji in this report was clearly presented in the Methods section as well as in the technical appendix. For example, the country-specific parameter values are presented in details in the “scale-up functions” section of the Supplement and the model stratifications by age and diabetes status are clearly introduced in the main text (lines 85-88).
- In Figure 2 in the main text, the color legend should be displayed graphically rather than textually to facilitate the association of the curves to different scenarios. In addition, the color palette makes many scenarios hardly distinguishable (e.g. 2, 5 and 6), also considering that the curves are very close to each other.
Figure 2 has been updated to improve readability.